# The economic burden of multimorbidity: Protocol for a systematic review

**Amrit Banstola** *, **Nana Anokye, Subhash Pokhrel**

Department of Health Sciences, Brunel University London, Uxbridge, Middlesex, United Kingdom

* amrit.banstola@brunel.ac.uk

## Abstract

Multimorbidity, also known as multiple long-term conditions, leads to higher healthcare utilisation, including hospitalisation, readmission, and polypharmacy, as well as a financial burden to families, society, and nations. Despite some progress, the economic burden of multimorbidity remains poorly understood. This paper outlines a protocol for a systematic review that aims to identify and synthesise comprehensive evidence on the economic burden of multimorbidity, considering various definitions and measurements of multimorbidity, including their implications for future cost-of-illness analyses. The review will include studies involving people of all ages with multimorbidity without any restriction on location and setting. Cost-of-illness studies or studies that examined economic burden including model-based studies will be included, and economic evaluation studies will be excluded. Databases including Scopus (that includes PubMed/MEDLINE), Web of Science, CINAHL Plus, PsycINFO, NHS EED (including the HTA database), and the Cost-Effectiveness Analysis Registry, will be searched until March 2024. The risk of bias within included studies will be independently assessed by two authors using appropriate checklists. A narrative synthesis of the main characteristics and results, by definitions and measurements of multimorbidity, will be conducted. The total economic burden of multimorbidity will be reported as mean annual costs per patient and disaggregated based on counts of diseases, disease clusters, and weighted indices. The results of this review will provide valuable insights for researchers into the key cost components and areas that require further investigation in order to improve the rigour of future studies on the economic burden of multimorbidity. Additionally, these findings will broaden our understanding of the economic impact of multimorbidity, inform us about the costs of inaction, and guide decision-making regarding resource allocation and cost-effective interventions. The systematic review's results will be submitted to a peer-reviewed journal, presented at conferences, and shared via an online webinar for discussion.

## Introduction

Multimorbidity, also known as multiple long-term conditions, is a growing public health problem worldwide. It refers to the co-occurrence of two or more chronic conditions in the same

**Data Availability Statement:** No datasets were generated or analysed during the current study. All relevant data from this study will be made available upon study completion.

**Funding:** The author(s) received no specific funding for this work. This study is part of the first author's PhD project and that the first author is sponsored by the Department of Health Sciences at Brunel University London.

**Competing interests:** The authors have declared that no competing interests exist.

individual. A recent systematic review and meta-analysis of 193 studies from both high-income countries (HIC) and low- and middle-income countries (LMIC) found that the pooled estimate of multimorbidity prevalence was 42.4% (95% CI 38.9% to 46.0%) [1]. The review also found that older adults and those with multiple conditions had a significantly higher prevalence of multimorbidity. Although it is more prevalent among older adults, it can affect people of all age groups, including children and young people [1, 2]. Evidence also shows that multimorbidity is common among people with mental health disorders and those from lower socioeconomic positions [3]. Studies have shown that multimorbidity is linked to poorer health outcomes, such as a higher risk of mortality and poorer quality of life [4]. It leads to higher healthcare utilization, including hospitalization, readmission, and polypharmacy, as well as a financial burden to families, society, and nations [4–11].

The variable definitions and measurements of multimorbidity have presented a significant challenge to research, as they complicate the comparison, interpretation, and synthesis of data, thus limiting the true burden of multimorbidity [12]. The lack of consensus on the definition and measurement of multimorbidity has also hindered the prevention and treatment of multimorbidity, including policymaking [13]. The definitions and measurements of multimorbidity are based on the simple count or number of conditions (e.g., two or more conditions), choice of conditions (e.g., by types and names of conditions), and even a holistic concept that includes learning disability, sensory impairment, alcohol and substance misuse [14]. Moreover, various weighted indices have been used to define and measure multimorbidity, such as the Charlson Index, the Cumulative Illness Rating Scale (CIRS), the Adjusted Clinical Groups (ACG) System, and the Chronic Disease Score [15–17].

The existing healthcare systems globally have a single-disease focus, aiming to treat specific chronic conditions. However, the rising burden of multimorbidity means that these systems would struggle to manage patients with multiple long-term conditions and increase the demand for scarce healthcare resources. To address the demand for healthcare and support decision-making, it is important to make a judgment about costs. Few studies have summarized the costs of multimorbidity [5–8, 11], and those that have are limited in scope and therefore do not provide a comprehensive estimate of the economic burden of multimorbidity. The earliest systematic review by Lehnert et al. [8] explored the relationship between multiple chronic conditions, healthcare utilization, and healthcare costs but was restricted to studies of older adults only. This review considered articles published from 1992 to 2009 and presented the results based on the count of diseases only, without examining the costs of specific disease combinations. Similarly, like Lehnert et al., the systematic review by Wang et al. [5] also presented the results based on the count of diseases only, without examining the costs of specific disease combinations. The review included studies published from 2000 to October 2016, which is now over six years old, and there might have been several studies within this time that explored the economic impact of multimorbidity.

One systematic review only considered the out-of-pocket spending on medicines by patients with non-communicable disease multimorbidity, thereby excluding infectious diseases and other important cost components such as inpatient and outpatient costs [7]. Another systematic review assessed the healthcare costs associated with multimorbidity by disease combinations and healthcare components, but was limited to UK studies only [11]. The most recent systematic review and meta-analyses examined the costs of specified sets of disease combinations, including comorbidity, thereby excluding studies not specifying other combinations of chronic conditions [6]. This systematic review also did not examine the costs based on the count of diseases and therefore had to exclude twenty-four studies with unspecified disease combinations. It is important to note that multimorbidity is different from comorbidity, which is the coexistence of additional diseases in relation to one disease of interest known as

an index disease in an individual [13]. Other reviews are expert reviews about the impacts of multimorbidity on healthcare costs and resource utilisation [9, 10], and do not meet the criteria for systematic reviews. As they are not systematic reviews, the evidence on the economic burden of multimorbidity that they present is not comprehensive.

Overall, the current state of knowledge regarding the economic implications of multimorbidity is characterized by fragmented evidence, limited scope, outdated information, and methodological limitations. To bridge these gaps and limitations in existing research, there is a clear need for a systematic review focused on the economic burden of multimorbidity. This systematic review will address these limitations by comprehensively synthesising existing research, encompassing a wide array of disease combinations, considering various cost components, and providing a nuanced understanding of how multimorbidity impacts healthcare costs. Aligned with the Academy of Medical Sciences' (AMS) research priorities [13], this review will provide evidence that has potential to inform healthcare policy, resource allocation, and decision-making in addressing the challenges posed by multimorbidity. The AMS has outlined multimorbidity research priorities, including providing a comprehensive estimate of the economic burden of multimorbidity, exploring the social and economic benefits of prevention and multimorbidity management, and assessing the costs of inaction [13].

### Research questions

The purpose of this systematic review is to answer the following research question: "What is the comprehensive estimate of the economic burden of multimorbidity in people of all ages, considering the various definitions and measurements of multimorbidity, and how does this cost vary by disease count, disease cluster, weighted indices, and across countries?"

### Aims and objectives

The aims of this systematic review are to identify and synthesise comprehensive evidence on the economic burden of multimorbidity, considering various definitions and measurements of multimorbidity, including implications for future cost-of-illness analyses.

The specific objectives of the review are to:

- Identify and synthesise evidence on the economic burden of multimorbidity, with an emphasis on how costs differ based on:

  a. the number/counts of diseases

  b. disease clusters/ combinations

  c. weighted indices

- Critically appraise the methodological and reporting quality of the included studies.

- Provide recommendations for improving the rigour of future studies on the economic burden of multimorbidity.

### Materials and methods

The Preferred Reporting Items for Systematic Reviews and Meta-Analyses Protocols (PRISMA-P) 2015 checklist [18] was used to write this protocol. The PRISMA-P checklist is available in S1 Checklist. The protocol was registered with the International Prospective Register of Systematic Reviews (PROSPERO) on 15 March 2023 (registration number CRD42023407338) [19]. The PRISMA 2020 checklist [20] will be used when writing the full systematic review.

### Inclusion and exclusion criteria

A detailed inclusion and exclusion criteria is shown in Table 1.

### Databases to be searched

The following electronic databases will be searched:

- Scopus (that includes PubMed/MEDLINE, 1788 onwards), Web of Science (Core Collection, 1970 onwards), CINAHL Plus (EBSCOhost, 1937 onwards), PsycINFO (EBSCOhost, 1872 onwards), NHS EED including HTA database (Centre for Reviews and Dissemination, until March 2015), and the Cost-Effectiveness Analysis (CEA) Registry (Tufts Medical Centre, 1976 onwards).

References collated by the 'International Research Community on Multimorbidity' (https://www.gla.ac.uk/schools/healthwellbeing/research/generalpractice/internationalmultimorbidity) will be screened for additional publications. The reference lists of eligible studies will also be

**Table 1. Inclusion and exclusion criteria for reviewed studies.**

| Category | Inclusion criteria | Exclusion criteria |
|---|---|---|
| Population | People of all ages with multimorbidity | None |
| Intervention | Not applicable | Not applicable |
| Comparison | Not applicable | Not applicable |
| Outcomes | • Costs (direct medical, direct non-medical costs, indirect costs such as lost productivity)<br>• Out-of-pocket expenses/ health expenditure | • Studies reporting incremental costs only<br>• Studies that did not report costs |
| Study design/type | • Observational studies (cross-sectional, cohort, case-control)<br>• Costing/cost-of-illness studies or economic burden studies including model-based studies | • Non-human research<br>• Abstract/conference proceeding<br>• Literature reviews or meta-analyses<br>• Trials<br>• Economic evaluation studies which compares two or more interventions (e.g. cost-effectiveness analysis, cost utility analysis, cost-benefit analysis)<br>• Study protocols<br>• Qualitative research<br>• Case series |
| Study location and setting | No restrictions on location (studies from all countries will be included e.g. low- and middle-income countries, high-income countries) and setting (community, primary care, care home, hospital providing both specialist and non-specialist service). | None |
| Data sources or data collection methods used | Patient self-report, physician reports (based on clinical examinations), medical records, administrative databases ('coded databases' or 'routine data') | Qualitative interviews |
| Multimorbidity definition | • Reporting any definition of multimorbidity.<br>• Studies that used the term multimorbidity without any explanation of how it is defined or justifying their underlying definition of it. | Studies with single chronic condition |
| Multimorbidity measurement | • Based on count/number of conditions or weighted indicies or types of conditions<br>• Weighted indices/measures such as various versions of the Charlson comorbidity index (CCI), the Cumulative Illness Rating Scale (CIRS), the Index of Coexistent Disease (ICED), the Adjusted Clinical Groups (ACG) System and the Duke Severity of Illness Check-list system, Elixhauser index, Kaplan indices, Seattle Index of Comorbidity, Self-administered comorbidity questionnaire, Shwartz Index, the Rx-risk comorbidity index, the Medication-Based Disease Burden Index (MDBI), Chronic Disease Score, and a medicines comorbidity index, multisource comorbidity score (MCS)<br>• Any disease combinations | Studies that did not measure multimorbidity. For example, studies with comorbidity (an index condition associated with one specific disease) |

screened. The search will be restricted to papers written in the English and published from inception to March 2024.

## Search strategy

The search strategy will include two concepts: 1) multimorbidity and 2) economic burden, and will be adapted from previously published systematic reviews [6, 12]. We will use a combination of Medical Subject Headings (MeSH) and text words related to multimorbidity and costs. For example, 'multimorbidity', 'comorbidity', 'polypharmacy', 'chronic illness', 'costs and cost analysis', 'cost of illness', 'financial stress', 'financial burden', 'out-of-pocket costs', 'health expenditure', 'catastrophic health expenditure', and 'healthcare cost'. A draft Scopus search strategy is included in S1 Appendix. Additionally, we will seek advice from the academic liaison librarian of the College of Health, Medicine, and Life Sciences at Brunel University to finalize the search strategy for this review.

## Study selection

After searching all databases for articles, the records of studies will be exported to RefWorks for deduplication and initial screening. After removing duplicates, two reviewers (AB and SP) will screen the titles and abstracts from a randomly selected 20% of studies to ensure consistency. Based on the interrater reliability [21] the first author (AB) will screen the remaining studies. All potentially relevant studies will have their full text retrieved and assessed against the inclusion and exclusion criteria. Any disagreements will be resolved through discussion and consensus with the third reviewer (NA). Studies that do not meet the inclusion criteria will be excluded, and the reason for exclusion will be provided in the S1 Appendix.

## Data extraction and management

A structured data extraction form will be created using Microsoft Excel. The data extracted will include, but is not limited to, the following: study details (author, year of publication), study characteristics (study design, patient characteristics, sample size, location, and setting), disease characteristics (definition and measurement of multimorbidity, disease conditions), outcomes, costs (perspective, resource use, costing year, currency, and sources of cost data), results, authors' conclusions, limitations of the study, and future research identified by the authors of included studies. The form will be piloted before use, and modifications will be made where necessary. Two reviewers (AB and SP) will independently extract data from a randomly selected 20% of included studies to ensure consistency. Based on the interrater reliability [21] the first author (AB) will extract data from the remaining studies. Any uncertainties will be resolved through discussion and consensus with the review team.

## Risk of bias assessment

To assess the risk of bias within included studies, the methodological quality of studies will be assessed by using a well-established appraisal checklist appropriate to the study design. For instance, the quality of observational studies such as case-control and cohort studies will be assessed using the Newcastle-Ottawa Scale (NOS) [22]. The NOS will evaluate the quality of studies based on three categories: a) population selection, b) comparability of the population chosen on the basis of design or analysis, and c) exposure or outcome. The standard NOS will be adapted as in previous systematic reviews [6, 7]. The quality of cost-of-illness studies will be assessed using the Larg and Moss checklist [23]. This checklist will also assess the potential biases in three broad categories: a) related to the analytical framework (perspective chosen,

epidemiological approach), b) methodology and data (identifying, measuring, and valuing resource uses and productivity losses), and c) analysis (including uncertainties) and reporting of the findings. Quality assessment will aid the interpretation of the analysis, but it will not determine exclusion. The risk of bias assessment will apply at the study level rather than the outcome level. The two reviewers (AB and SP) will independently assess the risk of bias within included studies after extracting information from a randomly selected 20% of studies to ensure consistency. Based on the interrater reliability [21] the first author (AB) will then assess the quality of the remaining studies. The review team will discuss and reach a consensus on any uncertainties.

## Data analysis and synthesis

We anticipate that a network/meta-analysis will not be possible due to the expected heterogeneity of findings (in terms of patients, study design, and costing approach) and inadequate detail provided by the included studies to conduct such an analysis. We will therefore, provide a narrative synthesis of the main characteristics and results of the studies included in this review. The summary will be organized according to the following: a description of the studies, including study design, study population, study settings, definition and measurement of multimorbidity, and key aspects related to costs (such as perspectives adopted, time horizon, resource use, type of costs, sources of cost data, and handling of uncertainty). The related conditions will be grouped together. The total economic burden of multimorbidity will be reported as mean annual costs per patient and disaggregated based on counts of diseases, disease clusters, and weighted indices. The findings will be presented as a percentage of the gross national income per capita, alongside data on health financing obtained from the WHO Health Financing dashboard [24]. The dashboard includes information on key aspects of health financing systems, such as funding sources, health spending, and other health financing indicators. In addition, we will explore how factors such as income inequality, as measured by the Gini coefficient, may influence the interpretation of our findings. This broader economic framework aims to provide a more nuanced understanding of the financial implications of multimorbidity for different populations.

The costs will be presented as direct medical costs (i.e., costs related to the use of resources due to disease treatment, such as inpatient stay and drugs), direct non-medical costs (i.e., costs related to the treatment process, such as travel, food, and informal care) indicating the economic burden on individual patients. Additionally, we will present any available information on indirect costs (i.e., costs incurred due to losses from the disease, such as loss of time and production) shedding light on burdens beyond the health care sector which will have broader societal implications. So, we will aim to provide a more comprehensive understanding of the economic impact beyond individual patients. We will conduct subgroup analysis where possible, based on sociodemographic factors such as age and sex, as well as clinical factors including counts of diseases, disease clusters, weighted indices, and body system involvement. Additionally, we will consider geographical location if relevant to the study context.

To facilitate comparability, all costs will be adjusted to 2022 UK Pounds using the GDP deflator index and purchasing power parity conversion rate. The analysis will be carried out using the Campbell and Cochrane Economics Methods Group (CCEMG)—Evidence for Policy and Practice Information and Coordinating Centre (EPPI-Centre) Cost Converter v.1.6, to ensure accuracy [25].

## Assessing the quality of evidence

The Grading of Recommendations Assessment, Development, and Evaluation (GRADE) approach is not appropriate for evaluating the quality of evidence for the outcomes in this

review [26]. This approach is designed to rate the quality of economic evidence, particularly evidence on the impacts of interventions on resource use and costs. However, as our review does not incorporate economic evaluation studies, the GRADE approach is not applicable for evaluating the quality of evidence in this context. Instead, we will utilise robust tools to assess the risk of bias in individual studies. This includes using the NOS for observational studies [22] and the Larg and Moss checklist for cost-of-illness studies [23], as mentioned in the 'Risk of bias assessment' section of this protocol. These tools are widely recognised and accepted for evaluating the quality of evidence in systematic reviews of observational and cost-of-illness studies, respectively.

## Preliminary timeframe

Table 2 provides a detailed breakdown of the estimated timeframes for each stage of the systematic review, including the selection of relevant studies, data extraction, quality assessment, and data synthesis. Based on these estimates, we anticipate that the entire review will be completed within a duration of six months.

## Discussion

To the best of our knowledge, this is the most comprehensive review on the economic burden of multimorbidity, using commonly accepted definitions and measurements. This includes assessment based on disease count, types of disease combinations, and weighted indices. By analysing the gross national income per capital alongside data on health financing from the WHO Health Financing Dashboard, this review will provide a deeper understanding of how these factors influence the economic burden of multimorbidity across diverse nations. This understanding also holds significance in facilitating the interpretation and generalization of the findings to various healthcare contexts.

### Implications for research, practice or policy

The primary contribution of this review is to provide evidence that can help explore how the costs of multimorbidity vary across disease clusters and countries. Additionally, the review will identify the key cost components that would be helpful in estimating the future cost-of-illness analysis of multimorbidity. Furthermore, the review's outcomes will provide valuable insights for researchers regarding areas that require further investigation and opportunities to improve the rigour of future studies on the economic burden of multimorbidity. By considering the full range of the economic burden of multimorbidity, this review aims to compare the costs across various definitions and measurements of multimorbidity. The findings from this review could

**Table 2. Timeline of the study.**

| Stage of review | Duration (weeks) |
|---|:---:|
| Develop a search strategy | 2 |
| Conduct a comprehensive search from relevant databases | 2 |
| Piloting of the study selection process | 1 |
| Screen studies for inclusion against eligibility criteria | 2 |
| Data extraction | 4 |
| Risk of bias (quality) assessment | 1 |
| Data analysis and synthesis | 4 |
| Write up the review (manuscript writing) | 7 |
| Dissemination (manuscript submission and presentation) | 2–4 |

be useful in advocating for the benefits of preventing and managing multimorbidity. In other words, the review's findings are crucial in expanding our understanding of the economic burden of multimorbidity. These insights can serve as an advocacy tool to strengthen disease prevention programmes and guide informed decision-making on resource allocation and health service organisation through the design and implementation of cost-effective interventions.

## Strengths and limitations of the study design

This systematic review has several strengths. Firstly, it has a clear set of inclusion and exclusion criteria. Secondly, a comprehensive search strategy, which will include a combination of keywords from previous systematic reviews, alongside guidance from a specialist librarian, will mitigate the risk of omitting relevant studies. Thirdly, a well-established quality assessment tool, tailored to the study design, will be utilised to appraise the included studies for relevance, reliability, validity, and applicability. Lastly, the review will adhere to PRISMA-P 2015 guidelines to ensure clarity and transparency in reporting. However, the review will be limited to English-language studies due to limited resources, possibly introducing publication and language biases. Nonetheless, the authors anticipate that many relevant studies are published and available in English. The expected heterogeneity of findings in terms of patients, study design, and costing approach, may preclude the feasibility of conducting network/meta-analysis. We will exclude full economic evaluation studies from our systematic review because they often only report incremental costs related to specific interventions or treatments for multimorbidity, rather than the full economic burden. These studies may lack sufficient detail on cost sources, methods, and components, particularly when focused on outcomes or cost-effectiveness ratios, limiting transparency, reproducibility, and generalizability of cost estimates across studies.

## Dissemination plans

The results of this systematic review will be submitted to a high-quality, peer-reviewed open access journal. The findings will also be presented at relevant academic conferences and stakeholders' meetings. An online webinar will be organized to present the findings, allowing for questions and discussions. The targeted audience for the webinar will primarily include healthcare professionals, policymakers, researchers, and other stakeholders involved in multiple long-term conditions.

## Amendments to the study protocol

There are no plans to make any amendments to the systematic review during the review process. However, in the event that any changes are made to the original protocol, the authors will clearly document them (providing the date of each amendment, describing the change with the rationale) and reflect them in both the PROSPERO and the published review.

## Supporting information

**S1 Checklist. PRISMA-P 2015 checklist.**
(DOC)

**S1 Appendix. A draft SCOPUS search strategy.**
(DOCX)

## Acknowledgments

The authors would like to thank the Brunel University library for their support in finalising the search strategy.

## Author Contributions

**Conceptualization:** Amrit Banstola, Nana Anokye, Subhash Pokhrel.

**Data curation:** Amrit Banstola.

**Formal analysis:** Amrit Banstola.

**Investigation:** Amrit Banstola.

**Methodology:** Amrit Banstola, Nana Anokye, Subhash Pokhrel.

**Supervision:** Nana Anokye, Subhash Pokhrel.

**Validation:** Amrit Banstola.

**Writing – original draft:** Amrit Banstola.

**Writing – review & editing:** Amrit Banstola, Nana Anokye, Subhash Pokhrel.

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
