## [Decision Letter · Decision Letter 0]

13 Jul 2023

PONE-D-23-09002The economic burden of multimorbidity: protocol for a systematic reviewPLOS ONE

Dear Dr. Banstola,

Thank you for submitting your manuscript to PLOS ONE. After careful consideration, we feel that it has merit but does not fully meet PLOS ONE’s publication criteria as it currently stands. Therefore, we invite you to submit a revised version of the manuscript that addresses the points raised during the review process.

We look forward to receiving your revised manuscript.

Kind regards,

Achyut Raj Pandey, MPH

Academic Editor

PLOS ONE

Journal Requirements:

“The author(s) received no specific funding for this work. This study is part of the first author's PhD project and that the first author is sponsored by the Department of Health Sciences at Brunel University London.”

Reviewers' comments:

Reviewer's Responses to Questions

**Comments to the Author**

1. Does the manuscript provide a valid rationale for the proposed study, with clearly identified and justified research questions?

Reviewer #1: Yes

Reviewer #2: Yes

2. Is the protocol technically sound and planned in a manner that will lead to a meaningful outcome and allow testing the stated hypotheses?

Reviewer #1: Yes

Reviewer #2: Yes

3. Is the methodology feasible and described in sufficient detail to allow the work to be replicable?

Reviewer #1: Yes

Reviewer #2: Yes

4. Have the authors described where all data underlying the findings will be made available when the study is complete?

Reviewer #1: Yes

Reviewer #2: Yes

5. Is the manuscript presented in an intelligible fashion and written in standard English?

Reviewer #1: Yes

Reviewer #2: Yes

6. Review Comments to the Author

You may also provide optional suggestions and comments to authors that they might find helpful in planning their study.

Reviewer #1: Reviewer’s report

Title: The economic burden of multi-morbidity: protocol for a systematic review

Version: 0

Date: 1st May 2023

Reviewer:

Reviewer's report:

Thank you for the authors for providing opportunity to review this protocol manuscript. This protocol aims to identify and synthesize the available evidence on economic burden of multi-morbidity. I believe this paper will be a crucial scholar contribution for decision makers for allocating resources to address multi-morbidity. I would provide following suggestions for further improvement of this paper. However, authors can use their discretion to agree or disagree with the feedback and consider the revision.

1. The authors have reviewed the available literature and the findings have been presented well. They have clearly presented the findings and limitations of each literature which is well appreciated. It would be better if the authors could state the rationale for performing this systematic review. If would be better if the readers could clearly understand what this systematic review will going to add in the literature that are already available.

2. The authors need to declare the source of financial and non-financial support for this systematic review. Although the authors have mentioned that they received non-financial support from the university library in finalizing the search strategy, the authors need to declare whether or not they have received any financial support for conducting this review.

3. I would suggest the authors to elaborate more on the discussion section.

Reviewer #2: Proposed Review intends to bring two important aspects together [morbidity and economic burden] which will be certainly contribute for the policy makers in evidence based decision making and open avenues for further specific studies.

The protocol states that searches will be done until March 2023- the timeline has already been passed- has the searches been already completed or still to be done. It needs to adjust accordingly.

The economic burden of multimorbidity very much depends on what kind of health financing system a particular country has. Although the paper states that it will review across the countries, but the one of the important factors that affects the economic burden is the health systems architect of that particular country. So, the systematic review should consider income level and health financing system of the country in concern for generalization of the findings from countries with varying income level and health financing system.

Although various indices have been included as the inclusion criteria for the review, it would be good to elaborate on what kind of weighted indices will be used as the methodology for economic burden analysis as stated in the objectives. Methodology and reporting quality could have major implication on the reported costs. The protocol is less explicit on how the ‘methodological and reporting quality’ [second objective] will interact on the analysis of the economic burden [first objective]. This is important to clarify in the protocol.

Economic burden could be to the patients versus to the overall health systems in itself which needs to be clarified in terms of scope of the study. If it is to focus on economic burden on individual patient, it has strong association with how the health financing system is designed which needs to be clarified.

In the background, it states that ‘’there might have been several studies 77 within this time that explored the economic impact of multimorbidity’’ indicating to review those papers. However, the paper has stated that it will exclude papers on ‘economic evaluation’. Please present it with logical linkage for consistency and clarity.

7. PLOS authors have the option to publish the peer review history of their article (what does this mean?). If published, this will include your full peer review and any attached files.

Reviewer #1: No

Reviewer #2: No

---

## [Author Response · Author response to Decision Letter 0]

19 Aug 2023

REVIEWER 1

Comment 1: The authors have reviewed the available literature and the findings have been presented well. They have clearly presented the findings and limitations of each literature which is well appreciated. It would be better if the authors could state the rationale for performing this systematic review. If would be better if the readers could clearly understand what this systematic review will going to add in the literature that are already available.

Response 1: Thank you for the comment. We acknowledge the importance of explicitly stating the rationale for conducting our systematic review and elucidating the unique contribution it makes to the existing literature. To address this concern, we have now improved the introduction section of our manuscript on page 5 by providing a rationale behind undertaking this systematic review. By highlighting the specific aspects that our systematic review will address, such as synthesizing diverse research, capturing various disease combinations, considering different cost components, and providing an in-depth understanding of the intricate relationship between multimorbidity and healthcare costs, we aim to clearly convey the unique value that our study adds to the existing body of literature.

Comment 2: The authors need to declare the source of financial and non-financial support for this systematic review. Although the authors have mentioned that they received non-financial support from the university library in finalizing the search strategy, the authors need to declare whether or not they have received any financial support for conducting this review.

Response 2: Thank you for your comment. We have clearly stated the funding information in the financial disclosure section of the submission system as required by the journal. The submission guidelines suggested the following:

“Do not include funding sources in the Acknowledgments or anywhere else in the manuscript file. Funding information should only be entered in the financial disclosure section of the submission system.”

For your information, we have stated the following financial disclosure in the submission system:

“The author(s) received no specific funding for this work. This study is part of the first author's PhD project and that the first author is sponsored by the Department of Health Sciences at Brunel University London.”

Comment 3. I would suggest the authors to elaborate more on the discussion section.

Response 3: Thank you for your suggestion to elaborate more on the discussion section. We have now added few sentences in the first paragraph of the Discussion section on page 14.

REVIEWER 2

Comment 1: The protocol states that searches will be done until March 2023- the timeline has already been passed- has the searches been already completed or still to be done. It needs to adjust accordingly.

Response 1: Thank you for highlighting this. We acknowledge that the stated timeline has now been passed. But we have not yet carried out the searches. We have therefore, amended this to August 2023 in the revised manuscript.

Comment 2: The economic burden of multimorbidity very much depends on what kind of health financing system a particular country has. Although the paper states that it will review across the countries, but the one of the important factors that affects the economic burden is the health systems architect of that particular country. So, the systematic review should consider income level and health financing system of the country in concern for generalization of the findings from countries with varying income level and health financing system.

Response 2: We appreciate your insightful comment to consider income level and health financing system of the country in the analysis. We agree that it is an important consideration when interpreting and generalizing our findings. In response to your valuable feedback, we added the following sentence in the ‘Data analysis and synthesis’ section on page 12.

“The findings will be presented as the percentage of the gross national income per capita alongside the health financing systems of the country.”

We have also added the following texts to highlight the importance of such analysis in the ‘Discussion’ section on page 14.

“By analysing the gross national income per capital and the health financing systems of each country, this review will provide a deeper understanding of how these factors influence the economic burden of multimorbidity across diverse nations. This understanding also holds significance in facilitating the interpretation and generalization of the findings to various healthcare contexts.”

Comment 3: Although various indices have been included as the inclusion criteria for the review, it would be good to elaborate on what kind of weighted indices will be used as the methodology for economic burden analysis as stated in the objectives. Methodology and reporting quality could have major implication on the reported costs. The protocol is less explicit on how the ‘methodological and reporting quality’ [second objective] will interact on the analysis of the economic burden [first objective]. This is important to clarify in the protocol.

Response 3: Thank you for your valuable comment concerning the use of weighted indices. To clarify, we plan to present the economic burden for each available weighted index, rather than adhering to a specific index as the sole methodology for economic burden analysis. This approach allows us to consider a variety of perspectives and capture the multidimensionality of economic burden assessments.

We appreciate that methodology and reporting quality could have major implication on the reported costs and that is what we are going to showcase through this review. In the ‘Risk of bias assessment’ section, we have outlined our strategy to evaluate the methodological quality of included studies using a well-established appraisal checklist appropriate to the study design. Additionally, we have emphasised that this quality assessment will aid the interpretation of the analysis, but it will not determine exclusion. We have amended the first sentence of ‘Data analysis and synthesis’ section to make it clear. The revised sentences now explicitly state our intention to provide a narrative synthesis due to the expected heterogeneity and insufficient detail in the included studies. 

Comment 4: Economic burden could be to the patients versus to the overall health systems in itself which needs to be clarified in terms of scope of the study. If it is to focus on economic burden on individual patient, it has strong association with how the health financing system is designed which needs to be clarified.

Response 4: Thank you for your valuable feedback. We completely agree that the economic burden can encompass both patients and overall health systems. To address this aspect, we have taken steps to clarify the scope of our study.

In the 'Data analysis and synthesis' section on page 12, we have explicitly stated that we will organize our findings based on the ‘perspectives adopted’, which could range from the patient perspective to the health system perspective. Furthermore, we have emphasized that we will report the ‘mean annual costs per patient’, indicating the economic burden on individual patients, particularly those related to direct medical costs and direct non-medical costs. Additionally, we will present any available information on indirect costs, shedding light on burdens beyond individual patients. In this manner, our study intends to capture the multifaceted nature of the economic burden.

We also appreciate your insight about the association between economic burden on individual patients and the design of the health financing system. We have taken your suggestion into account and made appropriate amendments to the manuscript (also as part of your comment #2).

Comment 5: In the background, it states that ‘’there might have been several studies 77 within this time that explored the economic impact of multimorbidity’’ indicating to review those papers. However, the paper has stated that it will exclude papers on ‘economic evaluation’. Please present it with logical linkage for consistency and clarity.

Response 5: Thank you for bringing up this point. We want to clarify that our review will specifically focus on the economic impact of multimorbidity. As you correctly noted, we will exclude papers categorised as ‘economic evaluation’ because they often involve comparing the costs and benefits of interventions and strategies aimed at preventing, managing or controlling multimorbidity. Thus, there inclusion is beyond the scope of this review. To make it clear, we have made the necessary adjustment on page 7. We added the phrase ‘which compares two or more interventions’ after ‘economic evaluation studies’ to explicitly highlight the distinction and reinforce our review’s exclusive emphasis on the economic impact of multimorbidity.

---

## [Decision Letter · Decision Letter 1]

28 Nov 2023

PONE-D-23-09002R1The economic burden of multimorbidity: protocol for a systematic reviewPLOS ONE

Dear Dr. Banstola,

Thank you for submitting your manuscript to PLOS ONE. After careful consideration, we feel that it has merit but does not fully meet PLOS ONE’s publication criteria as it currently stands. Therefore, we invite you to submit a revised version of the manuscript that addresses the points raised during the review process.

We look forward to receiving your revised manuscript.

Kind regards,

Filipe Prazeres, MD, MSc, Ph.D.

Academic Editor

PLOS ONE

Journal Requirements:

Reviewers' comments:

Reviewer's Responses to Questions

**Comments to the Author**

1. Does the manuscript provide a valid rationale for the proposed study, with clearly identified and justified research questions?

Reviewer #3: Yes

Reviewer #4: Yes

Reviewer #5: Yes

2. Is the protocol technically sound and planned in a manner that will lead to a meaningful outcome and allow testing the stated hypotheses?

Reviewer #3: Yes

Reviewer #4: Yes

Reviewer #5: Yes

3. Is the methodology feasible and described in sufficient detail to allow the work to be replicable?

Reviewer #3: Yes

Reviewer #4: Yes

Reviewer #5: Yes

4. Have the authors described where all data underlying the findings will be made available when the study is complete?

Reviewer #3: Yes

Reviewer #4: Yes

Reviewer #5: No

5. Is the manuscript presented in an intelligible fashion and written in standard English?

Reviewer #3: Yes

Reviewer #4: Yes

Reviewer #5: Yes

6. Review Comments to the Author

You may also provide optional suggestions and comments to authors that they might find helpful in planning their study.

Reviewer #3: This is a well-designed protocol study. But, since comorbidity includes a wide range of diseases, it seems that not limiting the study to a range of diseases (based on the global burden of disease or...), challenges the validity of the study.

However, some recommendations that may be helpful in improving the study design were included in the text of the article as comments.

page 6: Types of costs in the health sector include direct medical, direct non medical and indirect cost. Therefore, the cost as an inclusion criterion includes the three types of costs mentioned

page 7- Some full economic evaluation studies report all cost in detail, Why are they among the exclusion criteria?

page 11- subgroup analysis did not mentioned. it help to better understanding and comparing different condition?

Reviewer #4: I read your protocol thoroughly. This is an interesting systematic review protocol. It is written quite well and there are not many comments on it from my side at the moment. I would like to provide a couple of supportive comments to make the protocol more robust.

1. In Introduction section, line 102, you have included AMS in brackets, not sure why. Probably AMS shouldn’t be in brackets, please check it.

2. Please revise the inclusion of studies published date within November 2023 as it is already middle of the November (lines 147/8), as August has already gone.

3. In the section ‘Assessing the Quality of Evidence’ (lines 224-230), you have talked only about GRADE, which you said not suitable for this review. If there any other suitable tool, you can talk about that to assess the quality of evidence. If not remind the readers about your RoB tools for ‘quality assessment’ of papers and write down how these tools will be helpful to make sure your synthesised evidence is robust.

Other contents look good. Best wishes for your review. I am looking forward to reading your final systematic review report.

Reviewer #5: Feedback

Thank-you for the opportunity to review the presented manuscripts entitled “The economic burden of multimorbidity: protocol for a systematic review”.

The study aims to investigate an important subject-it stands to add valuable health economic data/information in the field of public health and inform key policy decision making discussions aimed at addressing global/country specific disease burden response.

The authors presented technically sound methods/approach for answering the study question. Authors applied conventional standard (E.g., PRISMA -P tool) to inform items to present/address in a systematic review protocol.

Recommendation: Protocol to be considered for publication once the below comments (categorized as non-major) have been addressed.

Specific comments:

Introduction

Line 44: multiple instead of larger number

Line 63: It is difficult to interpret the "The single chronic disease" Please expand.

Risk of bias assessment

Line 182: The risk bias assessment is fairly articulated. Please confirm if these s happen at the outcome or study level or both.

Data Analysis and synthesis

Line 210-211: Whilst I appreciate the reporting of the findings by gross national income per capita , please note that this measure may not necessarily reflect the countries/nationals financial capacity to absorb the financial burden posed by multi-morbidity. The interpretation should be in the context of other economic indices e.g., Gini coefficient ranking. Countries with high income in-equality may still present with a high GD per capita yet a majority of the households/individuals would be living below poverty line. This can be masked when GD per capita is interpreted in-silos.

Line 217: Indirect costs will shed light on economic burden beyond the health care provider, specifically direct and indirect cost provide societal costs.

Assessing the quality of evidence

Line 224-230: Whilst the grading of recommendation Assessment Development and Evaluation may not be appropriate - Please describe how this study will assess the strength of body of evidence that shall be presented.

Discussion

Line 244: Can the authors specify how they intend to analyse the countries' health financing systems. Does this mean there will be a separate analysis/assessment of the respective HF systems functioning /performance. This part is not clear. Please expand for clarity

Implications for research, practice or policy

Line 295: The study is not assessing costs of inaction. However, understanding the magnitude of multi-morbidity can be an advocacy tools for strengthening disease prevention programs.

Dissemination plans

Line 279: Who will be your targeted audience for the webinar?

General:

The authors did not articulate where/how data underlying the findings will be made available

7. PLOS authors have the option to publish the peer review history of their article (what does this mean?). If published, this will include your full peer review and any attached files.

Reviewer #3: No

Reviewer #4: No

Reviewer #5: No

---

## [Author Response · Author response to Decision Letter 1]

22 Feb 2024

Respond to reviewers comments have been attached as a separate file.

---

## [Decision Letter · Decision Letter 2]

18 Mar 2024

The economic burden of multimorbidity: protocol for a systematic review

PONE-D-23-09002R2

Dear Dr. Banstola,

We’re pleased to inform you that your manuscript has been judged scientifically suitable for publication and will be formally accepted for publication once it meets all outstanding technical requirements.

Kind regards,

Filipe Prazeres, MD, MSc, Ph.D.

Academic Editor

PLOS ONE

Additional Editor Comments (optional):

Reviewers' comments:

Reviewer's Responses to Questions

**Comments to the Author**

1. Does the manuscript provide a valid rationale for the proposed study, with clearly identified and justified research questions?

Reviewer #4: Yes

Reviewer #5: Yes

2. Is the protocol technically sound and planned in a manner that will lead to a meaningful outcome and allow testing the stated hypotheses?

Reviewer #4: Yes

Reviewer #5: Yes

3. Is the methodology feasible and described in sufficient detail to allow the work to be replicable?

Reviewer #4: Yes

Reviewer #5: Yes

4. Have the authors described where all data underlying the findings will be made available when the study is complete?

Reviewer #4: Yes

Reviewer #5: Yes

5. Is the manuscript presented in an intelligible fashion and written in standard English?

Reviewer #4: Yes

Reviewer #5: Yes

6. Review Comments to the Author

You may also provide optional suggestions and comments to authors that they might find helpful in planning their study.

Reviewer #4: I read the manuscript again. The revised manuscript looks good to me. I do not have any further comments on the manuscript.

Reviewer #5: The Author have done due diligence to address comments provided. As such the Proposed method and processes is quite robust and feasible. I highly recommend that the editor consider the manuscript for publication.

7. PLOS authors have the option to publish the peer review history of their article (what does this mean?). If published, this will include your full peer review and any attached files.

Reviewer #4: No

Reviewer #5: **Yes: **Dr Cebisile Ngcamphalala
